# CLASS INTERFERENCE OF DEEP NEURAL NETWORKS

## ABSTRACT

Recognizing and telling similar objects apart is even hard for human beings. In this paper, we show that there is a phenomenon of *class interference* with all deep neural networks. Class interference represents the *learning difficulty in data* and it constitutes the largest percentage of generalization errors by deep networks. To understand class interference, we propose cross-class tests, class ego directions and interference models. We show how to use these definitions to study minima flatness and class interference of a trained model. We also show how to detect class interference during training through label dancing pattern and class dancing notes.

## 1    INTRODUCTION

Deep neural networks are very successful for classification (LeCun et al., 2015; Goodfellow et al., 2016) and sequential decision making (Mnih et al., 2015; Silver et al., 2016). However, there lacks a good understanding of why they work well and where is the bottleneck. For example, it is well known that larger learning rates and smaller batch sizes can train models that generalize better. Keskar et al. (2016) found that large batch sizes lead to models that look sharp around the minima. According to Hochreiter & Schmidhuber (1997), flat minima generalize better because of the minimum-description-length principle: low-complexity networks generalize well in practice.

However, some works have different opinions about this matter (Kawaguchi et al., 2017; Dinh et al., 2017; Li et al., 2018). Dinh et al. (2017) showed that sharp minima can also generalize well and a flat minimum can always be constructed from a sharp one by exploiting inherent geometric symmetry for ReLU based deep nets. Li et al. (2018) presented an experiment in which small batch minimizer is considerably sharper but it still generalizes better than large batch minimizer by turning on weight decay. Large batch training with good generalization also exists in literature (De et al., 2017; Goyal et al., 2017). By adjusting the number of iterations, Hoffer et al. (2017) showed there is no generalization gap between small batch and large batch training.

These works greatly helped understand the generalization of deep networks better. However, it still remains largely mythical. In this paper, we show there is an important phenomenon of deep neural networks, in which certain classes pose a great challenge for classifiers to tell them apart at test time, causing *class interference*.

Popular methods of understanding the generalization of deep neural networks are based on minima flatness, usually by visualizing the loss using the interpolation between two models (Goodfellow et al., 2015; Keskar et al., 2016; Im et al., 2016; Jastrzebski et al., 2017; Draxler et al., 2018; Li et al., 2018; Lucas et al., 2021; Vlaar & Frankle, 2022; Doknic & Möller, 2022). Just plotting the losses during training is not enough to understand generalization. Linearly interpolating between the initial model and the final trained model provides more information on the minima.

A basic finding in this regard is the monotonic property: as the interpolation approaches the final model, loss decreases monotonically (Goodfellow et al., 2015). Lucas et al. (2021) gave a deeper study of the monotonic property on the sufficient conditions as well as counter-examples where it does not hold. Vlaar & Frankle (2022) showed that certain hidden layers are more sensitive to the initial model, and the shape of the linear path is not indicative of the generalization performance of the final model. (Li et al., 2018) explored visualizing using two random directions and showed that it is important to normalize the filter. However, taking random directions produces stochastic loss contours. It is problematic when we compare models. We take a *deterministic* approach and

study the loss function in the space of *class ego directions*, following which parameter update can minimize the training loss for individual classes.

The contributions of this paper are as follows.

- Using a metric called CCTM that evaluates class interference on a test set, we show that class interference is the major source of generalization error for deep network classifiers. We show that class interference has a *symmetry pattern*. In particular, deep models have a similar amount of trouble in telling "*class A objects are not class B*", and "*B objects are not A*".

- To understand class interference, we introduce the definitions of class ego directions and interference models.

- In the class ego spaces, small learning rates can lead to extremely sharp minima, while learning rate annealing leads to minima that are located at large lowlands, in terrains that are much bigger than the flat minima previously discovered for big learning rates.

- The loss shapes in class ego spaces are indicative of interference. Classes that share similar loss shapes in other class ego spaces are likely to interfere.

- We show that class interference can also be observed in training. In particular, it can be detected from a special pattern called *label dancing*, which can be further understood better by plotting the *dancing notes* during training. Dancing notes show interesting interference between classes. For example, a surprise is that we found FROG interferes CAT for good reasons in the CIFAR-10 data set.

## 2    CLASS INTERFERENCE

### 2.1    GENERALIZATION TESTS AND THE CLASS INTERFERENCE PHENOMENON

Let $c_1$ and $c_2$ be class labels. We use the following *cross-class test of generalization*, which is the percentage of $c_2$ predictions for the $c_1$ objects in the test set:

$$CCTM(c_1, c_2) = \frac{\# \text{ predicting as } c_2}{\# \text{total } c_1 \text{ objects}},$$

Note this test being an accuracy or error metric depends on whether the two classes are the same or not. Calculating the measure for all pairs of classes over the test set gives a matrix. We refer to this measure the CCT matrix, and simply the *CCTM* for short. CCTM extends the confusion matrix in literature by a probability measure, which can be viewed as a combination of the true positive rates and false positive rates in a matrix format [1]. This extension facilitates a visualization of the generalization performance as a heat map.

Figure 1 shows the CCTM for VGG19 (Simonyan & Zisserman, 2015) and ResNet18 (He et al., 2015) on the CIFAR-10 (Krizhevsky et al., 2009) test set with a heat map. Models were trained with SGD (see Section 3 for the training details). From the map, we can see that the most significant generalization errors are from CAT and DOG for both models. This difficulty is not specific to models. It represents *class similarity and learning difficulty in data*. For example, in Table 1, the accuracies in the columns of CAT and DOG are significantly lower than the other columns for all the four deep models. It is also observable that class interference has a **symmetry pattern**: If a classifier has trouble in recognizing that $c_1$ objects are not class $c_2$, it will also have a hard time in ruling out class $c_1$ for $c_2$ objects. This can be observed from CAT and DOG in the plotted CCTM.

We call generalization difficulties of deep neural networks between classes like CAT and DOG the **class interference**. If $CCTM(c_1, c_2)$ is large, we say that **class $c_2$ interferes** $c_1$, or **class $c_1$ has interference from** $c_2$. Class interference happens when classes are just similar. In this case, cats and dogs are hard to recognize for humans as well, especially when the resolution of images is low. Examining only the test error would not reveal the class interference phenomenon because it is an overall measure of all classes. The classes have a much varied difference in their test accuracies. For example, in VGG19, the recall accuracy of CAT, i.e., $CCTM(CAT, CAT)$, is only about 84.5%

---

[1]See https://en.wikipedia.org/wiki/Sensitivity_and_specificity for example.

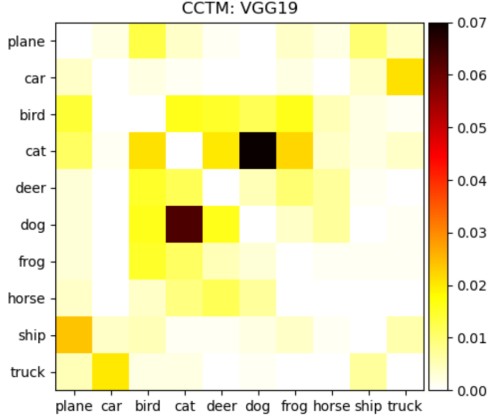 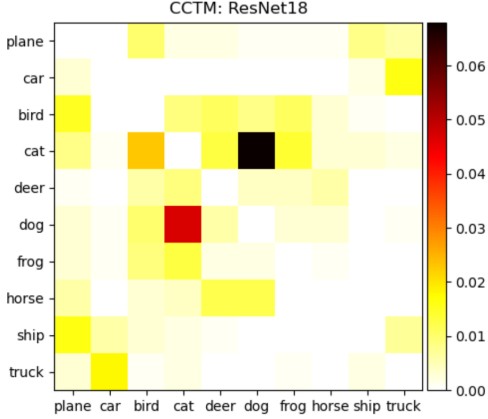

Figure 1: CCTM for VGG19 and ResNet18 on CIFAR-10 *test set*. There is severe ***class interference*** between CAT and DOG for both models. A darker color means a higher interference from the column class to the row class. The diagonal entries are not plotted for a clearer looking heat map. ResNet18 has less interference than VGG19 between CAT and DOG, indicated by: (1) there are fewer darker cells in the CAT row; and (2) the DOG-CAT cell is brighter (red vs. dark grey).

|           | plane | car  | bird | **cat** | deer | **dog** | frog | horse | ship | truck |
|-----------|-------|------|------|---------|------|---------|------|-------|------|-------|
| VGG19     | 96.0  | 96.6 | 91.9 | **84.5** | 94.7 | **89.0** | 96.0 | 96.4  | 95.2 | 96.3  |
| ResNet18  | 96.9  | 97.9 | 94.2 | **86.5** | 97.0 | **92.6** | 96.9 | 96.3  | 96.5 | 97.3  |
| GoogleNet | 96.4  | 96.7 | 93.8 | **86.9** | 97.3 | **92.0** | 97.2 | 95.5  | 96.5 | 96.6  |
| DLA       | 95.4  | 97.8 | 92.6 | **88.7** | 96.6 | **90.8** | 96.1 | 97.3  | 97.0 | 96.4  |

Table 1: Class recall accuracy (the diagonal entries of the CCTM) on CIFAR-10 *test set*. Classifying CAT and DOG is the bottleneck for all the models. Figure 1 shows that this is due to all the models suffer from the interference between CAT and DOG.

and DOG recall is about 89.0%. For the other classes the recall accuracy is much higher, e.g., CAR is 96.6%. As shown in Table 1, ResNet18 (He et al., 2015), GoogleNet (Szegedy et al., 2014) and DLA (Yu et al., 2017) have less class interference than VGG19 especially for CAT and DOG. For example, for ResNet18, $CCTM(CAT, CAT) = 86.5\%$ and $CCTM(DOG, DOG) = 92.6\%$.

## 2.2 DEFINITIONS

Let $w^*$ be a trained neural network model, e.g., VGG19 or ResNet18. We use the following definitions.

**Definition 1** (Interference Model Set). *Let $\mathcal{D}_c$ be the samples of class $c$ in a data set. Define the* **gradient of class c** *as the average gradient that is calculated on this set:*

$$\nabla f^{(c)}(w^*) \stackrel{def}{=} \frac{1}{|\mathcal{D}_c|} \sum_{(X,Y) \in \mathcal{D}_c} f'(w^*|X, Y).$$

*Accordingly, there are a set of class gradient directions for the model, $\{\nabla f^{(c)}(w^*)|c = 1, 2, \ldots, C\}$, where $C$ is the number of classes.*

*An* **ego model** *of class* **c** *is generated by using a scalar $\alpha_i$ in the class gradient direction:*

$$w_i^{(c)} = w^* - \alpha_i \nabla f^{(c)}(w^*).$$

*The set, $\mathcal{M}_c = \{w_i^{(c)}|i = 1, \ldots, m_c\}$, is the* **ego model set** *of class $c$. The set union, $\mathcal{M} = \cup_{c=1}^{C} \mathcal{M}_c$, is called the* **ego model set**.

This definition is based on that each $w_i^{(c)}$ is in the direction of minimizing the loss for predicting class $c$. Note that $w_i^{(c)}$ is a sample of "ego-centric" update, which minimizes the loss for class $c$

only. It therefore could cause an increase in the prediction errors for the other classes. We refer to the gradient of class $c$ as the **ego direction** of the class. Measuring the loss on the interference models thus tells the interference between classes.

**Definition 2** (Interference Space). *The model space* $\{w^{(c_1, c_2)} | (\theta_1, \theta_2) \in \Theta_1 \times \Theta_2\}$ *is called the* interference model space *of class $c_1$ and $c_2$, where an interference model is defined by*

$$w^{(c_1, c_2)} = w^* - \left( \theta_1 \nabla f^{(c_1)}(w^*) + \theta_2 \nabla f^{(c_2)}(w^*) \right).$$

*Define* $\mathcal{F}^{(c_1, c_2)} = \{f(w^{(c_1, c_2)}) | (\theta_1, \theta_2) \in \Theta_1 \times \Theta_2\}$, *which is the set of* interference losses *between the two classes. The 3D space,* $\Theta_1 \times \Theta_2 \times \mathcal{F}^{(c_1, c_2)}$, *is the* loss interference space, *or simply, the* interference space *(of class $c_1$ and class $c_2$ for model $w^*$).*

**Proposition 1.** *Any interference model is a convex combination of the ego models of the two classes.*

*Proof.* Let $w_i^{(c_1)}$ and $w_j^{(c_2)}$ be the ego model of class $c_1$ and $c_2$, respectively. According to their definition,

$$
\begin{aligned}
\lambda w_i^{(c_1)} + (1-\lambda) w_j^{(c_2)} &= \lambda w^* - \lambda \alpha_i \nabla f^{(c_1)}(w^*) + (1-\lambda) w^* - (1-\lambda) \alpha_j \nabla f^{(c_2)}(w^*) \\
&= w^* - \left( \lambda \alpha_i \nabla f^{(c_1)}(w^*) + (1-\lambda) \alpha_j \nabla f^{(c_2)}(w^*) \right) \\
&= w^{(c_1, c_2)},
\end{aligned}
$$

where setting $\theta_1 = \lambda \alpha_i$ and $\theta_2 = (1-\lambda)\alpha_j$ finishes the proof. $\square$

## 3 MINIMA: FLAT OR SHARP?

Our first experiment is to understand minima sharpness of learning rate using class ego directions. We will visualize in the interference space, $\Theta_1 \times \Theta_2 \times \mathcal{F}^{(c_1, c_2)}$. We use this loss: the *mistake rate* for the $z$-axis, which is the percentage of classification mistakes on the training set to give a loss measure in the same range across different plots. We visualize the loss of the models on the training set versus $\Theta_1 \times \Theta_2$, which is a uniform grid over $[-\sigma, \sigma] \times [-\sigma, \sigma]$, with 19 points in each direction. This gives 361 interference models between a given class pair. We use the ego directions of CAT-DOG (the most interfering class pair), TRUCK-CAR (with a significant level of interference), and HORSE-SHIP (with little interference). These plots measure how sensitive the training loss changes with respect to the directions that focus on optimizing specially for individual classes and the linear combinations of these directions. The center of each plot corresponds to the origin, $(\theta_1 = 0, \theta_2 = 0)$, at which a trained VGG19 or ResNet is located.

We study the models of VGG19 and ResNet18 trained with the following optimizer setups:

- **big-lr**. This optimizer uses a big learning rate, $0.01$. The momentum and weight decay are the same as the small-lr optimizer. Figure 2 shows for VGG19 (top row) and ResNet18 (bottom row).

- **small-lr**. This SGD optimizer uses a small learning rate $0.0001$. It also has a momentum (rate $0.9$) and a weight decay (rate $0.0005$).

- **anneal-lr**. Similar to the above optimizers, but with an even bigger (initial) learning rate. A big constant learning rate $0.1$ leads to oscillatory training loss and poor models. We thus decay it with an initial value of $0.1$ using a Cosine rule (Loshchilov & Hutter, 2016). This is the optimizer setup used to train the models in Section 2.1.

The input images are transformed with RandomCrop and RandomHorizontalFlip and normalization. The batch size is 128. The Cross Entropy loss is used. Each model is trained with 200 epochs. The test accuracies for the models are shown in the following table.

| VGG-small-lr | VGG-big-lr | VGG-anneal-lr | ResNet-small-lr | ResNet-big-lr | ResNet-anneal-lr |
|---|---|---|---|---|---|
| 84.99% | 88.76% | 93.87% | 86.88% | 91.31% | 95.15% |

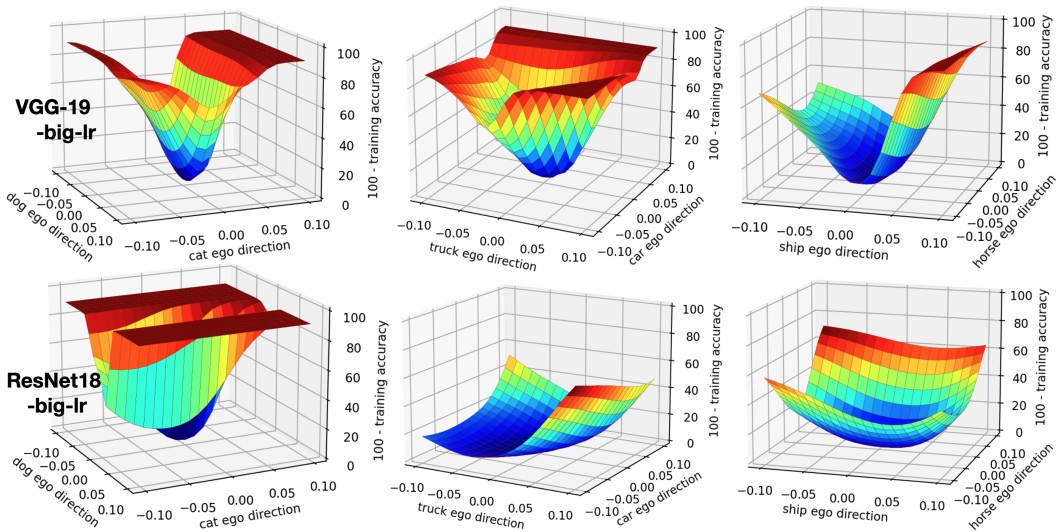

Figure 2: Loss visualized in the class ego directions for models optimized with big-lr. Model: VGG19 (top row); ResNet18 (bottom row). Each point stands for the loss of an interference model $w^{(c_1, c_2)}$.

This confirms that big learning rates generalize better than small ones as discovered by the community. Interestingly, the anneal learning rate leads to models that generalize even much better, for which there has been no explanation to the best of our knowledge.

Let's first take a look at VGG19 trained with big-lr, whose interference spaces are shown at the top row of Figure 2. The loss exhibits strong sharpness in the CAT-DOG ego visualization. From the minimum (the trained VGG19 at the center), a small step of optimizing the CAT predictions easily deteriorates the loss, in particular the red flat plateau corresponds to an accuracy on the training set down to merely 10%. The loss change is extremely sensitive in the CAT ego direction. It is similarly sensitive in all directions except near the DOG ego direction, which looks still very sensitive. According to Proposition 1, any interference model in this space is a convex combination of a CAT ego model and a DOG ego model. This plot thus shows that the CAT ego is very influential even the weight of the DOG ego is large.

The visualizations in the CAR-TRUCK and HORSE-SHIP ego spaces show that the loss changes much less sensitively than for CAT and DOG when we update the model for the purpose of improving or even sacrificing the prediction accuracy of the four classes. However, close to the directions of TRUCK ego plus negative CAR ego, and negative TRUCK ego plus CAR ego, the loss also changes abruptly. If we cut the loss surface 135 degrees in the $x$-$y$ axis, we end up getting a minimum that looks sharp. On the other hand, a random cut likely renders a less sharp or even flat look of the minimum. The case of HORSE-SHIP is similar. Thus whether the minimum looks flat or sharp is dependent on how the loss contour is cut. Some care needs to be taken when we discuss minima sharpness, especially the space in which the loss is plotted. Most previous discussions on minima sharpness are based on the difference between an initial model and a trained model, or two random directions. Both methods have randomization effects and yet they get descent loss contours. While it is amazing, the reason why random cuts render reflective loss contours is unclear. Our guess is that most directions renders sharpness and sampling a random one is likely fine. However, when we compare the levels of sharpness between models, random cuts may not be accurate.

Figure 2 bottom row shows for ResNet18 optimized with the big-lr optimizer. The loss change near the minimum is also extremely sensitive in the CAT-DOG ego space. Interestingly, for ResNet18, the loss in the DOG ego direction is more sensitive than in the CAT direction. This seems a "transposed" effect of VGG19, because the influence of the DOG ego is stronger on the loss now. For both VGG19 and ResNet18, the loss visualized in the CAT-DOG ego space has a clear narrow valley structure near the minimum. This kind of loss functions are known to be very challenging for gradient descent, e.g., see the Rosenbrock function also known as the Banana function (Rosenbrock, 1960). In the

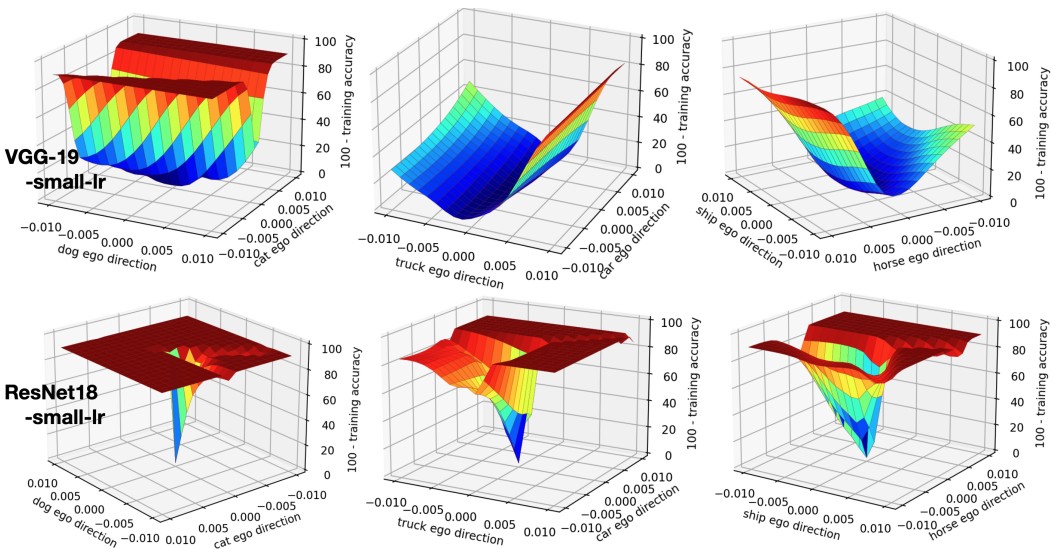

Figure 3: Loss visualized in the class ego directions for models optimized with small-lr. ResNet18 is a spiky (extremely sharp) minimum according to the visualization in the ego spaces, even though the area of the illustration is 1/100 of Figure 2 (big-lr).

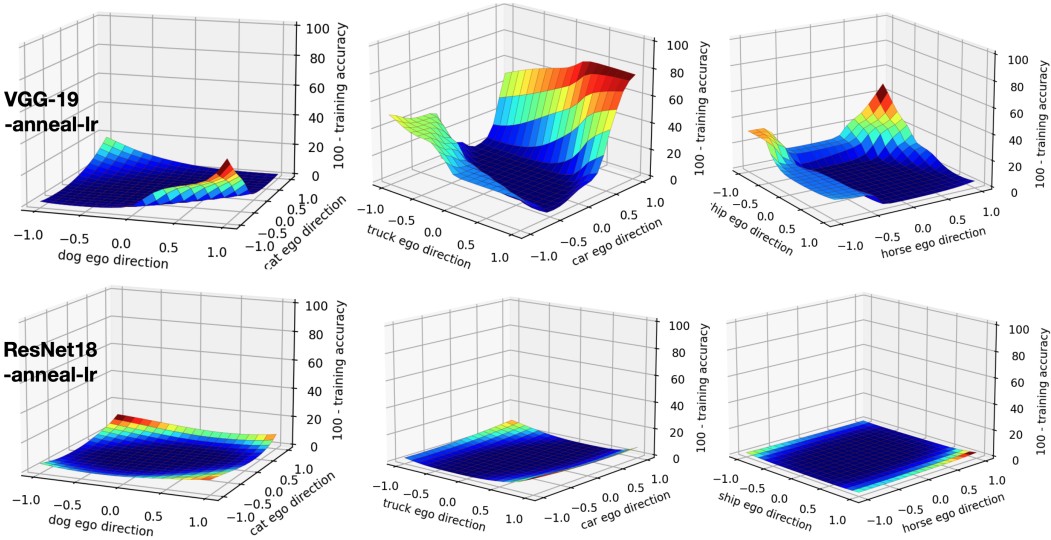

Figure 4: Loss visualized in the class ego directions for models optimized with anneal-lr. In this illustration area 100 times of Figure 2 (big-lr), both minima are located at a large flat lowland. The ResNet18 has an extremely large flat area.

CAR-TRUCK space, the loss of ResNet18 is much less curvy up than that of VGG19. In particular, for VGG19 it is sensitive in both the ego directions, while for ResNet18, only near the direction about 135 degrees ($x$-$y$ axis) it is sensitive. For VGG19, the SHIP direction has lots of sensitivity. For ResNet18, the HORSE direction instead is more sensitive.

Our results show the minima being ***flat or sharp*** is dependent on what spaces the loss is illustrated. We think a better way of discussing generalization is the ***area of flatness*** around the minima in critical directions. Our plots in different class ego spaces show that a minimum can be a flat minimum in certain visualization spaces (e.g., ResNet18 in the CAR-TRUCK ego space), while at the same time it can look very sharp in other spaces (e.g., ResNet18 in the CAT-DOG space).

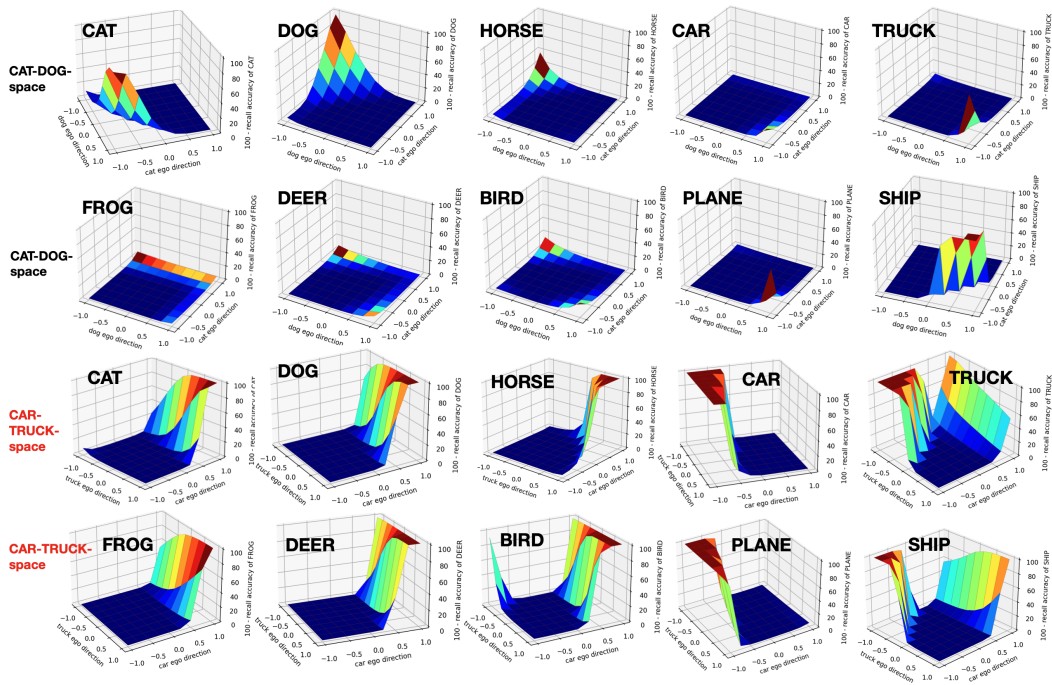

Figure 5: Class-wise losses. This shows all class prediction losses are more sensitive in the CAR-TRUCK space than in the CAT-DOG space. In particular, the CAR ego direction is very sensitive, which means all the other class prediction losses are greatly influenced by the movement in minimizing or maximizing (gradient ascent) the CAR loss. Model: VGG19. Optimizer: anneal-lr.

Figure 3 shows the small learning rate. This time ResNet18 is an extremely sharp minimum in all the three ego spaces. In a small area around the minimum in ego spaces, the loss changes dramatically. Beyond that small area, the loss is invariantly high (plateau). VGG19, instead, has a more smooth change of loss in a small area although in the CAT ego direction the loss changes abruptly too (which forms a cliff). This shows when the learning rate is small, the loss contour can be near non-smooth and sharp minima do not necessarily generalize worse (comparing to VGG19). This confirms the findings by Dinh et al. (2017) and (Li et al., 2018) that there exist models that are sharp minima and yet they still generalize well. In particular, ResNet18 has a better generalization than VGG19, 86.88% versus 84.99% in this case. Our results show that flat minima generalize better when the learning rate is well tuned (not too small). However, when the learning rate is small, the minima can be sharp and they can generalize even better than less sharp ones.

Finally, Figure 4 shows for the models optimized with learning rate annealing. These two models have superior generalization, with 93.87% for VGG19 and 95.15% for ResNet18. The visualization in the ego spaces show that the area of flatness is very large, especially ResNet18. Comparing to a fixed big learning rate, the models trained by annealing have a much higher level of in-sensitiveness to parameter changes in the class ego directions. Presumably, the big initial learning rate helps establish a larger flat area. This level of flatness has not been observed before, especially in previous experiments of learning rates. We may thus refer to minima located in a large flat terrain the ***lowland minima***.

## 4    ANALYZING CLASS INTERFERENCE

### 4.1    INTERFERENCE FROM ONE CLASS TO THE OTHERS

We also would like to understand the interference from one class to the others for a trained model. Figure 5 shows the interference of CAT, DOG, CAR and TRUCK to all the classes. First let's look at the CAT loss in the CAT-DOG space (first plot). It shows CAT loss increases in the cat ego direction,

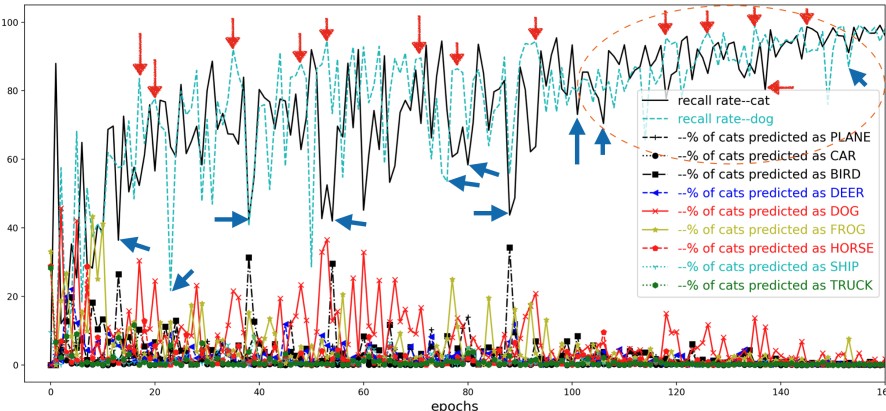

Figure 6: The CAT-DOG dance. Each red arrow is a moment that DOG interferes CAT, with a local high DOG recall rate (cyan dashed line) and a drop of the CAT recall rate (black line), and a local peak of mis-classified cats as DOG (red cross). Each blue arrow is a sample moment that both recall rates drop, in response to a high mis-classification rate of cats as a third class, e.g., BIRD (black rectangle). Model: VGG19. Optimizer: anneal-lr.

i.e., the gradient *ascent* direction, which is intuitive. It also shows the CAT loss increases most when we minimize the DOG loss. This is another verification that DOG interferes CAT. Interestingly, following the joint direction of gradient ascent directions to maximize the CAT loss and the DOG loss doesn't increase the CAT loss much. In the case of CAR loss in the CAR-TRUCK space, the situation is a little different. In particular, CAR loss increases significantly whether we follow the gradient descent or ascent direction of TRUCK as long as we move in the ascent direction of CAR. TRUCK loss is more complicated. The loss increases in the joint direction of ascent directions of CAR and TRUCK losses. In addition, TRUCK loss also increases if we follow the the descent direction of CAR. This means minimizing the CAR loss has the effect of increasing the TRUCK loss. This is also a sign that CAR and TRUCK interferes. For the other classes, their prediction losses respond more sensitively to the ego directions of CAR-TRUCK than those of CAT-DOG.

In the CAT-DOG space, CAR, TRUCK, PLANE, and SHIP all increase their losses in one same corner. HORSE and DEER losses both increase as we get closer to the corner where DOG loss increases; in addition, the increase of HORSE is more than DEER in this process.

In the CAR-TRUCK space, CAT, FROG, DOG, and DEER losses have very similar shapes. This suggests these losses increase in roughly the same directions in the CAR-TRUCK space. HORSE's loss shape is also similar to these four classes, but the similarity is less. CAR and PLANE losses have very similar shapes. TRUCK and SHIP losses have a similar wing-like structure too. CAR, PLANE, TRUCK and SHIP have similar loss shapes on the left side of the plots shown. These observations suggest that loss shapes in class ego spaces are indicative of interference. Classes that share similar loss shapes in other class ego spaces are likely to interfere. This is going to be discussed further in the next experiment.

## 4.2 CLASS INTERFERENCE IN TRAINING

The above experiments are for a trained model. We were wondering whether class interference can be observed in training. To study this, we plot the per-class training accuracy which is the recall rate for each class. Figure 6 shows for CAT and DOG. The two recall rates are both highly oscillatory, especially in the beginning stage of training. Importantly, there are many moments that one rate being high while the other being low at the same time, which we call ***label dance*** or CAT-DOG dance for this particular case. This dancing pattern is a strong indicator that CAT and DOG interfere. To further confirm this, we plot in the same figure the row of the CCTM for the training set that correspond to CAT, i.e., $CCTM(CAT, c)$, for each non-CAT class $c$, during the same training process. As the caption of the figure shows, a rise in the DOG recall rate is often caused by a high interference of DOG to CAT. After some (about 118) epochs, DOG interference dominates CAT

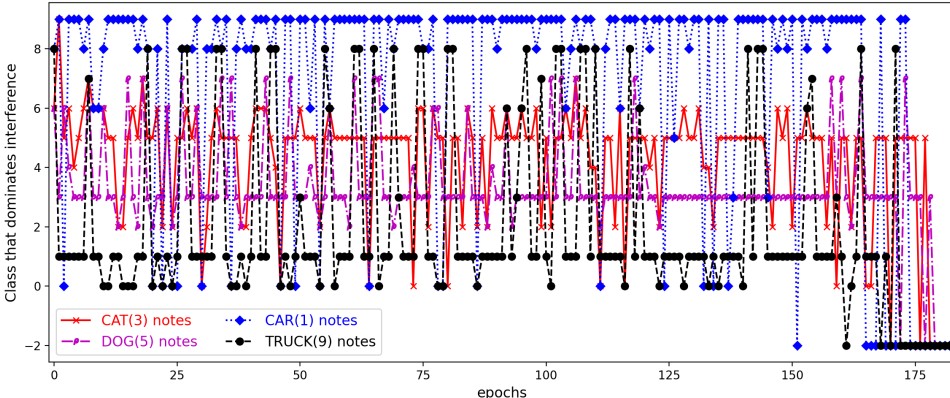

Figure 7: The dancing notes for CAT, DOG, CAR and TRUCK. A data point of the CAT notes curve is the most often false prediction at that epoch. Model: VGG19. Optimizer: anneal-lr.

predictions errors and eventually weeds out following the other classes. In this phase of training (as circled in the figure), the recall rates of CAT and DOG are highly symmetric to each other (horizontally), further indicating that DOG interference is the major source of error in predicting cats and vice versa.

Figure 7 plots the "argmax" operation of the CCTM for the rows corresponding to four classes at each epoch, excluding the diagonal part. The plot looks similar to music notes. So we term this plot "dancing notes". For DOG notes, there are many pink markers at the line $y = 3$, which is the class label corresponding to CAT. The stretched markers laying continuously is a clear sign of CAT interference to DOG. In the CAT notes, continual red crosses also persist at $y = 5$, which is the class label of DOG, showing interference of DOG to CAT. It also shows that CAT interference to DOG persists longer than the other way. For a better presentation of the results, we plot $y = -2$ if no class interferes more than $0.1\%$.

The notes of CAR (class label 1) and TRUCK (class label 9) show similar duration of interference, and it appears the interference from CAT to TRUCK seems to have a close strength to the other way around. It is also interesting to observe that both CAR and TRUCK have interference from class labels $y = 0$ and $y = 8$, which correspond to PLANE and SHIP. This is intuitive because these are all human made metallic crafts. It appears that the interference from PLANE to TRUCK is more often than to CAR, probably because trucks are bigger in size than cars.

CAT has interference from BIRD (2) given their similar fluffy looks. Surprisingly, FROG (6) also interferes CAT pretty often. We checked the CIFAR-10 images visually and it is probably because the images are mostly close looks of the objects; in this case cats have two pointy ears which are easily confused with frogs who have their eyes positioned atop. Besides CAT, DOG has interference from HORSE (7) and DEER (4) because they are all four-legged. It is interesting to observe that CAT, on the other hand, almost does not have interference from HORSE, with only two or three moments of interference out of 200 epochs. This means HORSE is very helpful to differentiate between CAT and DOG, which is the largest source of generalization error as we discussed in Section 2.1. DOG also has a little interference from BIRD (2) similar to CAT does.

## 5 CONCLUSION

This paper illustrates a phenomenon called class interference of deep neural networks. We show it is the bottleneck of classification, which represents learning difficulty in data. The proposed cross-class generalization tests, class ego directions, interference models and the study of class-wise losses in class ego directions provide a tool set for studying the generalization of trained deep neural networks. The study of *label dancing* via the *dancing notes* provides a method of detecting class interference during training. With the provided tools in these two dimensions, we hope this paper is useful to understand the generalization of deep nets, improve existing models and training methods, and understand the data better as well as the learning difficulty of recognition.

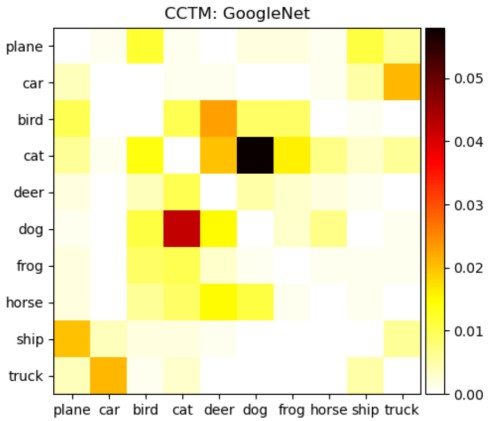 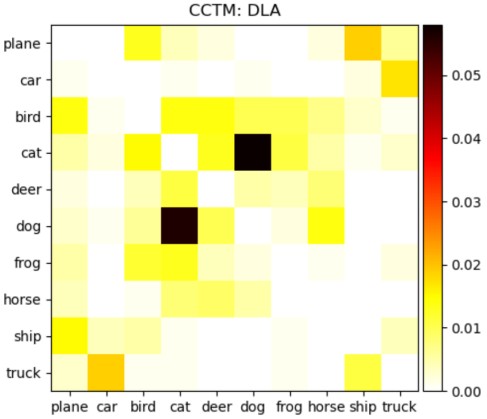

Figure 8: CCTM heatmaps for GoogleNet and DLA on CIFAR-10 test set. CAT and DOG still have the largest generalization errors among all the class pairs. GoogleNet wins DLA in the DOG class. However, it loses in the CAT class. See Table 1 for the recall rates of individual classes.

APPENDIX [2]

The CCTM heatmaps of GoogleNet and DLA are shown in Figure 8. The interference between CAT and DOG is similarly high (see Figure 1 for VGG19 and ResNet18).

Let us examine the CAR row in details this time. The CAR-TRUCK cell stands out. For GoogleNet, the color is a distinct Orange while other cells have very light colors. By looking at the colorbar, Orange color is about 0.025, which is 2.5%. In Table 1, GoogleNet's CAR recall is 96.7%. If we add them together, 96.7% + 2.5% = 99.2%. (For the TRUCK-CAR cell, it shows symmetry.) This means the majority of the generalization errors happen for predicting cars as TRUCK and vice versa. According to our experiments, CAR and TRUCK also interfere and we investigate their individual losses and illustrate their interference in training. Relevant discussions are Section 4.1 (Figure 5) and Section 4.2 (Figure 7).

For DLA, the color of CAR-TRUCK is much brighter than GoogleNet, which is more Yellow than Orange. Note the colorbar range of the two nets is the same and the colors across the two plots are comparable. This color is about 0.018 according to the colorbar. In Table 1, DLA's recall for CAR is 97.8%. We have 97.8% + 1.8% = 99.6%. This shows most of DLA's mistakes for cars happen for predicting them as TRUCK, similar to GoogleNet. We can also observe here that the CAR-TRUCK mistake of DLA (1.8%) is better than that of GoogleNet (2.5%). This leads to an overall better classification of cars for DLA (97.8%) than GoogleNet (96.7%).

> Reviewer question: If interference is something bad for classification (as you say "interference is the bottleneck"), then why PLANE/BIRD with lower interference (brighter colors) than CAR/TRUCK have better numbers in Table 1, e.g. for GoogleNet? PLANE/BIRD: 96.4/93.8 vs CAR/TRUCK 96.7/96.6?

We take a look at the raw CCTM data for GoogleNet, in particular the rows for PLANE, CAR, BIRD, and TRUCK. The data is shown in Table 2. So this confirms the recall rates in Table 1 are correct, in particular, PLANE/BIRD: 96.4%/93.8% vs. CAR/TRUCK 96.7%/96.6%.

Regarding why CAR/TRUCK have better recall rates than PLANE/BIRD, we can look at the CAR row first: All the numbers are low (0.00X or 0) except for the TRUCK column. This means GoogleNet rarely mistakes cars for classes other than TRUCK. Class interference is mainly for comparing the column classes for each row class. For the CAR row, TRUCK interefers it a lot, much more than the other classes.

---

[2]This section benefits from the discussions with one reviewer of ICLR 2023. It was added due to his suggestions.

| Class | PLANE | CAR | BIRD | CAT | DEER | DOG | FROG | HORSE | SHIP | TRUCK |
|-------|-------|-----|------|-----|------|-----|------|-------|------|-------|
| PLANE | 0.964 | 0.001 | 0.012 | 0.001 | 0. | 0.002 | 0.002 | 0.001 | 0.011 | 0.006 |
| CAR | 0.004 | 0.967 | 0. | 0.001 | 0.001 | 0. | 0. | 0.001 | 0.005 | 0.021 |
| BIRD | 0.01 | 0. | 0.938 | 0.01 | 0.023 | 0.009 | 0.009 | 0. | 0.001 | 0. |
| TRUCK | 0.004 | 0.021 | 0.001 | 0.003 | 0. | 0. | 0. | 0. | 0.005 | 0.966 |

Table 2: Sampled rows of GoogleNet for Figure 8.

For the PLANE row, (PLANE, BIRD) and (PLANE,SHIP) are both over 0.01. It makes sense because they both have sky background in the data set. (PLANE, TRUCK) is also a bit high, 0.006, because both are metallic.

That is, for the PLANE row, there are three other classes that interfere it, while for CAR/TRUCK, there is only one class that interferes it (let's say we use an interference threshold 0.005. Class B interferes class A if the cell (A, B) is bigger than 0.5%). Although the errors of (CAR, TRUCK) and (TRUCK, CAR) are high, the model does not make much other mistake for predicting cars and trucks. Thus their recall rates are high.

For the BIRD row, there are five classes with high interference to it too. Thus for PLANE and BIRD, there is also significant interference from other classes besides the most interfering class (the most dark color cell in a row). This leads to lower recall rates for PLANE/BIRD than for CAR/TRUCK.

This question and discussion shows that "many interfering classes for a class" is also bad for the class in addition to a single, strong "most interfering class". In short, "interference is the bottleneck" means the certain classes have strong interference from one or *multiple* other classes.

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
