# OpenReview forum: "Class Interference of Deep Networks"
_ICLR.cc/2023/Conference — Submitted to ICLR 2023_

### Official Review · Reviewer_aWAb · 2022-10-21

**Confidence:** 4
**Correctness:** 2
**Technical Novelty And Significance:** 1
**Empirical Novelty And Significance:** 2
**Recommendation:** 5

**Clarity, Quality, Novelty And Reproducibility:**

### Clarity

* The paper is sometimes confusing. The authors introduce a lot of metaphors that might be original, but confuse the reader, i.e. label dancing, dancing notes etc. I do not think these metaphors improve the paper, but rather the opposite.
* For the notation, do we need both $i$ and $j$ together with $c_1$ and $c_2$? Cam we just use one of them, e.g. in proposition 1?

### Quality

Some of the paper's claims are not justified.

* 'This difficulty is not specific to models. It represents class similarity and learning difficulty in data' I do not think this is shown. This also contradicts to the title of the paper 'Class interference of **neural networks**'.
* 'This [empirical results] confirms that big learning rates generalise better than small ones.' I think 'confirm' is a bit too strong. We can say that this supports/goes in line with, but no empirical result can confirm a hypothesis.

Page 5 compares the loss landscapes with the Rosenbrock function. I am not sure if we can compare the loss landscape of this paper with those since the optimisation takes the average gradient across all classes or across i.i.d. samples from the dataset, we do not update per-class.

I think that the Proposition 1 proof has a mistake, and that the Proposition itself is not true for all ego models of two classes: the proof proves the other direction of the implication. "**Proposition 1**: Any interference model is a convex combination of the ego models of the two classes." If this holds for any interference model, I pick $\theta_1=\lambda\alpha_i$, and I pick $\theta_2=\lambda\alpha_j$, which will not give me a convex combination unless $\lambda=1/2$. The proof seems to show that a convex combination of two ego-models is, in fact, a class interference model. Apart from that, it is not clear to me why we need Proposition 1. Why is knowing this important, what can we do with this knowledge?

### Novelty

* The CCTM metric looks very similar to the confusion matrix used throughout the classification literature (in fact, the CCTM matrix is a normalised confusion matrix).
* Task interference is a popular research direction in multi-task learning, with the goals of many multitask optimisers is somehow to alleviate the interference. However, recently, there has been a line of work showing that these multitask optimisers have a regularisation effect, and are not more effective than a simple regularised baseline (summing the gradients):
- Do Current Multi-Task Optimization Methods in Deep Learning Even Help?, NeurIPS 2022
- In Defense of the Unitary Scalarization for Deep Multi-Task Learning, NeurIPS 2022

The two works above have references to the most popular multitask optimisers as well.

### Reproducibility

The paper is quite scarce on the setting. It would be useful for the reader to have a more complete discussion of the exact training/testing hyper parameters in the appendix.

Apart from that, all the claims in the paper are based on the results of a single seed, and I believe that the plots might look differently if we run SGD with a different seed.


### Nits

* The ref to Rosenbrock might have a bibtex error. It says 'HoHo Rosenbrock' instead of 'H.H.Rosenbrock'.
* the Table on page 4 does not have a caption/title.

**Strength And Weaknesses:**

### Strengths

* The paper shows some interesting phenomena, e.g. the fact that lr annealing leads to flat minima for both of the architectures.
* I liked the introduction doing the overview of the related work and providing the context for the study.

### Weaknesses

Class interference is an intuitive phenomenon. However, whether this intuition aligns with optimisation challenges is unclear. It is also unclear what to do with this: yes, updating on one class only can make the other class deteriorate. What do we do with this? Should we even do something with this? If yes, why?

Apart from this, I don't think the paper demonstrates what it claimed to have demonstrated. For instance, looking at the conclusion:

* We show it [the class interference] is the bottleneck of classification <- I don't think the paper actually showed that
* ...we hope this paper is useful to understand the generalisation of deep nets <- it is unclear what exactly we understood from the fact that the classes interfere and that an updating one class leads to deterioration on the other.
* ...useful to ... improve existing models and training methods <- this would be an extremely useful application, but it is unclear how to use the proposed metrics to do so.



**Summary Of The Paper:**

The paper investigates 'class interference', i.e. how does update on one class affect the others. The work looks into the flatness of the minima for the converged models of two architectures (VGG and ResNet) trained with three different hyper parameter settings (smaller lr, larger lr and annealed lr). Finally, the authors look at how class interference evolves throughout training.

**Summary Of The Review:**

Given my comments above, I believe that the paper is not ready for a publication at ICLR. Apart from my criticisms above, I will put suggestions on how to improve the paper below:

* I think it would be extremely helpful to make a synthetic task with two overlapping gaussians to illustrate the authors' point. The gaussians are super easy to plot and very easy to vary the mean/std to get the desired effect.
* I know this is hard to do, but having a set of actionable insights will be extremely helpful for the researchers or practitioners. For instance, what should we do about the observed phenomenon?
* How, do you think, the irreducible error is related to class interference?

---

> ### Author Response · Authors · 2022-11-04
> **Could you elaborate? We don't understand your reviews because there are no informative backups**
>
> We show it [the class interference] is the bottleneck of classification <- I don't think the paper actually showed that.
>
> > Seriously? The paper didn't show this? We're very lost about this comment. Table 1 and the Figure 1 are for this purpose. CAT and DOG have the lowest recall rates. Can you tell us why this is not shown in details?
>
> Some of the paper's claims are not justified. 'This difficulty is not specific to models. It represents class similarity and learning difficulty in data' I do not think this is shown.
> >Again. Same as above. We're very confused by your brievity. We don't understand how comes this is not obvious from the paper.
>
> the use of CIFAR-10:
> > Please look at the top of our page, which addresses this.

---

> > ### Comment · Reviewer_aWAb · 2022-11-07
> > **response**
> >
> > > Seriously?
> >
> > I do not think that emotionally-charged response is a good way of constructively discussing the paper's merits/drawbacks.
> >
> > > The paper didn't show this? We're very lost about this comment. Table 1 and the Figure 1 are for this purpose. CAT and DOG have the lowest recall rates. Can you tell us why this is not shown in details?
> >
> > I do not think that showing interference of cat/dog on heatmaps and showing low recall is enough to justify the claim that the interference is the bottleneck of classification. Even if there is a correlation, it does not mean that interference *is* the bottleneck. Can we also see CCTM for GoogleNet/DLA from the table?
> >
> > > We don't understand how comes this is not obvious from the paper.
> >
> > Could you please, point me to specific paragraph in your paper or elaborate it here, why you think that the learning difficulty is not specific to the models, but to the data? Why cannot the interference be a result of both data properties as well as the model specification?

---

> > > ### Author Response · Authors · 2022-11-07
> > > **thanks for the reply.**
> > >
> > > We were confused by your brevity mostly. There was no emotional response except surprise in a technical way. Thank you for elaborating it now.
> > >
> > > I do not think that showing interference of cat/dog on heatmaps and showing low recall is enough to justify the claim that the interference is the bottleneck of classification. Even if there is a correlation, it does not mean that interference is the bottleneck. Can we also see CCTM for GoogleNet/DLA from the table?
> > >
> > > > Could we make sure that you already had a look at the following? (1) the captions of the Table 1, Figure 1. (2) the description texts for Figure 1 and Table1. If you have not, we can re-iterate a bit.
> > >
> > > >For CAT and DOG, all the models have very poor generalization for these two columns. For VGG19, the lowest recall rate happens at the CAT, which is 86%. The best recall is at TRUCK, which is 96%. That's 10% gap. For DOG, the recall is 90%, which is also very low comparing to other classes. Now, do we agree that CAT and DOG's recall rates are the bottleneck of the generalization for this dataset? This is the message of Table 1.
> > >
> > > >To see whether this is caused by the interference between the two classes, we need to combine Table 1 and Figure 1. Table 1 shows CAT and DOG are the bottleneck of the generalization. Figure 1 shows why and where does the test errors happen. Figure 1 shows that most CAT test errors are caued by wrong predictions of many cats as DOG. Vice versa is just the symmetry (the symmetry phenomenon described in the paper). That's exactly what we call the interference between CAT and DOG.
> > >
> > > >Hope this is clear for your comprehension on "the interference is the bottleneck of classification". This conclusion is perfectly illustrated, explained and defined by Table 1 and Figure 1. Please tell us what have caused an understanding of this difficult and we can improve.
> > >
> > > >The CCTM for the other models are similar and we just don't have space to show them. ResNet performs the best in the overall generalization anyway.  Other models are not needed to be included in the mainbody of the paper. If you're interested, we can generate these two figures and provide for you to have a look.

---

> > > > ### Comment · Reviewer_aWAb · 2022-11-08
> > > > **response**
> > > >
> > > > > Could we make sure that you already had a look at the following?
> > > >
> > > > Yes, I looked at the figures/tables and read the captions.
> > > >
> > > > > Now, do we agree that CAT and DOG's recall rates are the bottleneck of the generalization for this dataset?
> > > >
> > > > I don’t think so. Maybe we have a different understanding of the ‘bottleneck’ meaning. For me, it means something like ‘the reason for high classification errors’. While we can see that on the shown plots, there is a correlation, I don’t think this is enough to say that interference is the reason of low performance on these classes. In other words, yes, we see high error rates on cats/dogs, and we see that the network confuses the two, but can we say that this is the bottleneck and we cannot obtain lower error rates because of the cat/dog interference?
> > > >
> > > > > If you're interested, we can generate these two figures and provide for you to have a look.
> > > >
> > > > Could you, please, add the heatmaps for other models to the appendix?

---

> > > > > ### Author Response · Authors · 2022-11-08
> > > > > **Thanks for the discussion**
> > > > >
> > > > > the bottleneck of the generalization for this dataset:
> > > > >
> > > > > > This refers to the meaning by what's typed. Checking the meaning of "Bottleneck" in dictionary: a narrow section of road or a junction that impedes traffic flow. This refer to the particular class pairs that impede the generalization.
> > > > >
> > > > > For me, it means something like ‘the reason for high classification errors’
> > > > > >The reason is not the meaning here. However, absolutely the reason is important. The reason of high classification error is investigated by the remainder of the paper. Why does this happen? We investigate the minima sharpness, the CAT/DOG ego directions, and CAT/DOG loss shapes, and their CCTMs in training, etc.
> > > > >
> > > > > plot and attach the heatmaps for other models to the appendix:
> > > > > >Sure. We can do that.
> > > > >
> > > > > >Looking at your reasoning, it looks you have a different understanding of class interference. Class interference refers to class A objects are mis-classified by a model to Class B at testing time with a large rate. This is it and nothing about the deeper reasons. We will make this clearer in the revision. Wait. This is actually already in the paper. If you check this sentence, "If CCTM (c1, c2) is large, we say that class c2 interferes c1, or class c1 has interference from c2".
> > > > >
> > > > > >Class interference is a phenomenon (see abstract). For example, it refers to low recall rates of CAT and DOG whilst a high mis-classification of CATs as DOG; and verse versa. It's not about the reason why poor generalization between CAT and DOG happens.
> > > > >
> > > > > can we say that this is the bottleneck and we cannot obtain lower (Overal Test) error rates because of the cat/dog interference?
> > > > > >First please make sure adding the bracket is what you mean. If so, our answer is YES. Let's say T_cat is the total number of cats in the test set, and T_others is the total number of other testing objects. Let t_cat (t_others) be the number of correct CAT (Other) predictions. Then the overall testing accuracy is, %test = (t_cat + t_others)/(T_cat+T_others). If %cat = t_cat/T_cat decrease, then so does %test. This is because the denominator terms are constant.

---

> > > > > > ### Author Response · Authors · 2022-11-09
> > > > > > **CCTMs for GoogleNet and DLA are attached now to the revised paper**
> > > > > >
> > > > > > We just did experiments for GoogleNet and DLA. Their CCTM heatmaps are now attached to the appendix as requested.  Please let us know if you have more questions.
> > > > > >
> > > > > > We also did a second run of VGG19 and ResNet18, which produced similar heatmaps and recall rates. Fig 1 and Table 1 are also updated. Their numbers change slightly from the run used for submission.

---

> > > > > > > ### Comment · Reviewer_aWAb · 2022-11-09
> > > > > > > **thanks!**
> > > > > > >
> > > > > > > Thanks for the updates!
> > > > > > >
> > > > > > > How could you explain the fact that for GoogleNet, there is also high interference between `car` and `truck` compared to other classes except cat/dog, but the Table 1 numbers are high for these two classes? Similar thing can be said about DLA.

---

> > > > > > > > ### Author Response · Authors · 2022-11-09
> > > > > > > > **good question**
> > > > > > > >
> > > > > > > > if we look at Fig8 in the Appendix, and examine the CAR row, it shows the CAR-TRUCK cell stands out.
> > > > > > > >
> > > > > > > > For GoogleNet (left plot), the color is a distinct Orange while other cells have very light colors. Looking at the colorbar, Orange color is about 0.025, which is 2.5%. In Table 1, GoogleNet's CAR recall is 96.7%. If we add them together, 96.7% + 2.5% = 99.2%. For the TRUCK-CAR cell, it shows symmetry. This means the majority of the generalization errors happen for predicting CARs as TRUCK and vice versa. According to our experiments, CAR and TRUCK also interferes and we investigate their individual losses and illustrate their interference in training. Relevant discussions are Fig 5/Section 4.1 and Fig 7/4.2.
> > > > > > > >
> > > > > > > > For DLA, the color of CAR-TRUCK is much brighter than GoogleNet, which is more Yellow than Orange. Note the colorbar range of the two nets are the same and the colors across the two plots are comparable. This color is about 0.018 according to the colorbar. In Table 1, DLA's recall for CAR is 97.8%. We have 97.8% + 1.8%=99.6%. This shows most of DLA's mistakes for predicting CARs happen for predicting them as TRUCK, similar to GoogleNet. We can also observe here that CAR-TRUCK mistake of DLA (1.8%) is better than that of GoogleNet (2.5%).  This leads to an overall better classification of CARs for DLA (97.8%) than GoogleNet (96.7%).

---

> > > > > > > > > ### Author Response · Authors · 2022-11-09
> > > > > > > > > **we put this discussion in the paper**
> > > > > > > > >
> > > > > > > > > We feel this is a good discussion which can help future readers. Thus we put this discussion in the Appendix right below Fig8. It's easier to read the updated pdf because it is clearer to look at this alongside of the figure.
> > > > > > > > >
> > > > > > > > > Thanks.

---

> > > > > > > > > ### Comment · Reviewer_aWAb · 2022-11-10
> > > > > > > > > **response**
> > > > > > > > >
> > > > > > > > > Thanks for your reply.
> > > > > > > > >
> > > > > > > > > I mean something different with my question. What I meant is, if interference is something bad for classification (as you say 'interference is the bottleneck') why plane/bird with lower interference (brighter colors) than car/truck have better numbers in Table 1, e.g. for GoogleNet? plane/bird: 96.4/93.8 vs car/truck 96.7/96.6?

---

> > > > > > > > > > ### Author Response · Authors · 2022-11-10
> > > > > > > > > > **cute observation.**
> > > > > > > > > >
> > > > > > > > > > We take a look at the raw data for GoogleNet, in particular the CCTM rows for Plane, CAR, BIRD, and TRUCK.
> > > > > > > > > >
> > > > > > > > > > | Class  | PLANE | CAR | BIRD | CAT |  DEER | DOG | FROG | HORSE | SHIP | TRUCK |
> > > > > > > > > > | ----------- | ----------- |----------- | ----------- |----------- | ----------- |----------- | ----------- |----------- | ----------- |----------- |
> > > > > > > > > > | PLANE | 0.964 | 0.001 | 0.012 | 0.001 | 0. |  0.002 | 0.002 | 0.001 | 0.011 | 0.006 |
> > > > > > > > > > | CAR | 0.004 | 0.967 | 0. | 0.001 | 0.001 | 0. |  0. |  0.001 | 0.005 | 0.021 |
> > > > > > > > > > | BIRD | 0.01 | 0. | 0.938 | 0.01 |  0.023 | 0.009 | 0.009 | 0. | 0.001 | 0.  |
> > > > > > > > > > | TRUCK | 0.004 | 0.021 | 0.001 | 0.003 | 0.    | 0.    | 0.    | 0.    | 0.005 | 0.966 |
> > > > > > > > > >
> > > > > > > > > > So this confirms the data (recall rates) in Table 1 is correct, plane/bird: 96.4/93.8 vs car/truck 96.7/96.6.
> > > > > > > > > >
> > > > > > > > > > Regarding why CAR/TRUCK have better recall rates than PLANE/BIRD, we can look at the CAR row first: All the numbers are low (0.00X or 0) except for the TRUCK column. This means for cars, GoogleNet rarely mistakes CARs for classes other than TRUCK.
> > > > > > > > > >
> > > > > > > > > > Class interference is mainly for comparing the column classes for each row class. For the CAR row, TRUCK intereferes it a lot, much more than the other classes.
> > > > > > > > > >
> > > > > > > > > > For PLANE row, PLANE-BIRD and PLANE-SHIP are both over 0.01. It makes sense because they all have sky background in the dataset. PLANE-TRUCK is also a bit high, 0.006, because both are metallic.
> > > > > > > > > >
> > > > > > > > > > That is, for the PLANE row, there are three other classes that interfere it, while for CAR/TRUCK, there is only one class that interferes with it (let's say we use an interference threshold 0.005. Class B interferes class A if the cell (A, B) is bigger than 0.5\%). Although the errors of (CAR, TRUCK) and (TRUCK, CAR) are high, the model does not make much other mistakes for predicting CARs and TRUCKs. Thus their recall is high.
> > > > > > > > > >
> > > > > > > > > > (For BIRD row, there are five classes with high interference to it too.)
> > > > > > > > > >
> > > > > > > > > > Instead, for PLANE and BIRD rows, there are also significant interferences from other classes besides the most interferencing class (dark color cell). This leads to lower recall rates for them than for CAR and TRUCK.
> > > > > > > > > >
> > > > > > > > > > It looks our discussion of class interference could benefit from adding "many interferencing classes" in addition to "the most interferencing class".
> > > > > > > > > >
> > > > > > > > > > In short, ``interference is the bottleneck'' means the certain classes have strong interference from one or *multiple* other classes.
> > > > > > > > > >
> > > > > > > > > > Thanks. We really benefit from this discussion. We updated the Appendix again.

---

> > > > > > > > > > > ### Comment · Reviewer_aWAb · 2022-11-11
> > > > > > > > > > > **response**
> > > > > > > > > > >
> > > > > > > > > > > I appreciate that our discussion results in paper updates, thanks!
> > > > > > > > > > >
> > > > > > > > > > > I have one last question. What does the knowledge about class interference give us? Any particular insights? What should we do with this knowledge? Intuitively it's clear that if the network makes mistakes it confuses one class for another.  But why knowing this is interesting/useful?

---

> > > > > > > > > > > > ### Author Response · Authors · 2022-11-11
> > > > > > > > > > > > **lessons the authors learned**
> > > > > > > > > > > >
> > > > > > > > > > > > The most important lesson we learned is that the *data* itself causes large generalization errors in certain class predictions for deep models. Some models may do better than others for certain classes, while others may do better for another group of classes. They however all have a poor generalization for some tricky class pairs (e.g., CAT and DOG in this case). Classe pairs have different  levels of learning difficulty and generalization errors, and interference between class pairs is really a big challenge to all deep models.
> > > > > > > > > > > >
> > > > > > > > > > > > This message perhaps indicates that understanding this interference pattern in the data is important. This can help us develop better optimizers and deep models by examining their strength in handling class interference, instead of just relying on the overall test accuracy, which does not give much information and insights.
> > > > > > > > > > > >
> > > > > > > > > > > > This paper is devoted to gaining an understanding this phenomenon from the training data and during training. Below summarizes interesting observations:
> > > > > > > > > > > >
> > > > > > > > > > > > 1. The interference patterns tend to be symmetric. CAT recall error and DOG recall error are both high for all models, and their values are the closest among class errors. The symmetry indicates that the models may struggle a lot between CAT and DOG predictions during training for cats and dogs.
> > > > > > > > > > > >
> > > > > > > > > > > > 2. This is verified by Fig 6 and Fig 7. A lower recall of CAT very often corresponds to a high recall of DOG. If there is certain boundary between them, this means at these moments, the boundary was shifted closer to CAT.
> > > > > > > > > > > >
> > > > > > > > > > > > 3. There are also other contributions made by the paper, e.g., the summarization in the introduction. This thread of contributions is in the line of understanding the generalization (overall test accuracy; Fig 2, 3, 4) and interference (class-wise test accuracy; Fig 5) by visualizing the minima, for which we have a few interesting findings. This includes the extremely sharp minima for small learning rates, the much bigger and flatter terrain around the minima due to the annealed learning rate than the big learning rate. Previous research only compares small and big learning rates.
> > > > > > > > > > > >
> > > > > > > > > > > > 4. Oh, for item 3, we used the class ego directions, which is also very novel. This is related to the interesting question, in what space should we visualize the minima? Following Goodfellow et. al., 2015, most works in this regard use the interpolation models between the initial model and the final model (or two final models, small batch and large batch used in the flat minima 2016 paper by Keska et. al.). Li. et. al. 2018 plots the loss contour in a space of two (normalized) random directions. This renders stochastic loss contours and minima plots. When comparing different models, this randomness is not good and can lead to different comparison conclusions in different runs. We think we need to find derministic directions. Then we found the class ego directions. That's how the paper was developed.

---

> > > > > > > > > > > > > ### Comment · Reviewer_aWAb · 2022-11-14
> > > > > > > > > > > > > **response**
> > > > > > > > > > > > >
> > > > > > > > > > > > > Thanks for the response. I will continue discussing the paper with the other reviewers.

---

> ### Author Response · Authors · 2022-11-04
> **metaphors**
>
> This reviewer seems to be bothered by "metaphors".
>
> Yes. There are a few important phenoemons. The metaphors are used for better understanding. "Label dancing" and "dancing pattern (notes)" are the two most important ones. They are very well defined. Even their naming is explained in detail. The figures and captions, labeling in the figures all render a clear description of the phenomenons.
>
> If one doesn't read these in detail, could we just give a decision of rejection? Because it suffices to tell your decision about the paper. It is very confusing to read the reviews which show the paper wasn't read much, and key and important concepts and contributions are not understood due to insufficient reading. If you feel there is certain part that explains the concept is not clear, please point out with, specifics and backups of your opinion.
>
> It is always welcome to point problems of the paper (isn't that what we are volunteering for as reviewers?) . However, please give the backups when you do.

---

### Official Review · Reviewer_3tyq · 2022-10-22

**Confidence:** 3
**Correctness:** 2
**Technical Novelty And Significance:** 3
**Empirical Novelty And Significance:** 2
**Recommendation:** 5

**Clarity, Quality, Novelty And Reproducibility:**

Clarity:
The paper defines class interference related concepts in Section 2, and then discuss two ways to use class interference in Section 3 and 4. The paper is well organized.

Quality:
The motivation/application for studying class interference was not discussed in depth, for example, I wasn't sure how looking at the label dance during training can be helpful for training better neural networks. Only related work about flat minima were discussed, and other related work such as learning difficulty of data was not discussed. It would make the paper better if the relationship between related work is discussed further. I would also like to suggest exploring other datasets to see if the findings are general, since the paper currently only studies CIFAR-10. It would be interesting if the symmetry pattern only holds for certain datasets (such as image datasets).

Novelty:
The CCTM is identical to the confusion matrix (except the normalization part), which is already used heavily to study the performance of machine learning classifiers. The idea of using the interference space/label dance seems to be novel.

Reproducibility:
For reproducibility, the code was not included in the submission. The (optional) reproducibility statement was not included in the paper.

**Details Of Ethics Concerns:**

I do not have any ethical concerns.

**Strength And Weaknesses:**

Strengths:
- Studying the minima flatness/sharpness and learning difficulty in data are important topics, and class interference provides us with insights about the datasets. We can have a better understanding of CIFAR-10 using the tools provided in the paper.

Weaknesses:
- Section 2 explains how the difficulty of data based on CCTM is not specific to models, but it seems to be model dependent, since the results depend on the model used (although we can see similar trends between the different models).
- One of the main contribution is the class interference measure based on the proposed CCTM, but CCTM may not be novel, since it seems to be equivalent to the confusion matrix (or the normalized version of it). Confusion matrix is already used heavily to study the class-wise performance of deep neural network classifiers, especially in the industry.
- The discussions about "dance" seems interesting, but I'm not sure if I understood the phenomenon correctly. It would be helpful if there are more discussions on why if the training recall of one class rises, the other class's training recall will decrease. Furthermore, it is hard to visually confirm if this is happening, so would be nice to see the correlation value between these two time-series data.
- Currently the paper only studies the CIFAR-10 dataset, but it would be interesting to see if the same results, e.g., symmetry pattern, arise for other datasets as well.
- From the perspective of learning difficulty in data, there are many papers recently working on this, such as: "Deep Learning Through the Lens of Example Difficulty" (NeurIPS 2021), "Estimating Example Difficulty Using Variance of Gradients" (CVPR 2022), "Understanding Dataset Difficulty with V-Usable Information" (ICML 2022). There are also papers such as "Evaluating State-of-the-Art Classification Models Against Bayes Optimality" (NeurIPS 2021) that study class-wise difficulty (Appendix B.1 shows how CAT and DOG are the most difficult classes for CIFAR-10, which is consistent with the experimental results in the paper under review). It would be interesting to see discussions about related work.

**Summary Of The Paper:**

This paper studies deep neural network through the lens of "class interference". Class interference corresponds to difficulty for a pair of classes, where one class and another class are hard to distinguish for the neural network. More specifically the cross-class test of generalization matrix (CCTM) is used to measure the class interference between classes. Class interference becomes severe when classes are conceptually similar, as given by the example of Cats vs. Dogs. The paper further defines the ego model of each class, which is an gradient-updated model based on the average gradient of samples for only that class. Based on this idea, the paper defines an interference space, which is used to study trained deep neural networks and during training of deep neural networks.

**Summary Of The Review:**

Although the paper works on an interesting problem with new and existing tools, overall, due to the reasons I wrote in the previous sections, I am feeling that in its current form, it is below the acceptance threshold. I hope the authors can incorporate the comments from the reviewers to make the paper better.

---

> ### Author Response · Authors · 2022-11-04
> **similar trends between the different models**
>
> it seems to be model dependent, since the results depend on the model used (although we can see similar trends between the different models).
> >>Of course there are little differences between models.  Can we phrase reviews positively? for the same sentence you wrote -> although it appears to be model-dependent (with small variantions), we can see similar trends between the different models. It's hard to communicate and discuss ideas and research when reviews are written with an assumped negativeness.

---

> > ### Author Response · Authors · 2022-11-04
> > **dance**
> >
> > >>confusion matrix: It's not a measure. It's important to do the recall rates, false positive rates, which is the CCTM. Industry used a lot. Then could we say our extended and refined defition is an important contribution to the research community? could we show some positiveness in reviewing papers? I review papers at neurips, icml, aaai too. I was an Outstanding reviewer for icml 21. Please support good research with novel ideas and show positiveness.
> >
> > The discussions about "dance" seems interesting, but I'm not sure if I understood the phenomenon correctly.
> >
> > >>The experiment, figures: motivation, setup, and results interpretation are very detailed for understanding this concept. Perhaps spend more time understanding this interesting pattern will greatly help you.

---

> ### Author Response · Authors · 2022-11-05
> **Strengh**
>
> wow... The summary of Strength: your list only covers our Fig 1 and Table 1.
>
> This just just one page of our paper. And Fig 1 and Table 1 all use the test data set.
>
> All the remaining experiments and figures use the training set to gain an undersanding of why observations in Fig 1 and Table 1 happened.

---

### Official Review · Reviewer_6BAT · 2022-10-26

**Confidence:** 4
**Correctness:** 2
**Technical Novelty And Significance:** 2
**Empirical Novelty And Significance:** 2
**Recommendation:** 1

**Clarity, Quality, Novelty And Reproducibility:**

Clarity & Quality: The paper lacks clarity in terms of plotting and writing.
 - The motivation for class interference is not explained clearly.
 - The theoretical part seems superfluous.
 - The plot in Figures 6 & 7 has too many line plots, making it hard to distinguish.
 - The experiment part goes without a statement of settings.

Novelty: I'm not familiar with optimization literature and this paper seems novel to me.

Reproducibility: The code was not included in the submission. Also, the hyperparameters of models are not available.

**Strength And Weaknesses:**

Strength:
1. The visualization validates the claims.

Weakness:
1. In section 1, I'm confused about how the interpolation monotonicity is related to class interference. The motivation is not quite clear. Besides, it seems to be related to the form in Proposition 1, yet this proposition is not used or explained in this paper.
2. CCTM seems to be an extension for the confusion matrix when the number of classes is over 2.  Therefore, I'm concerned about the novelty of this metric.
3. CCTM can't explain the difference between the failed generalization of an overfit model and an underfit model, which by intuition has very different loss landscapes. For example, in an underfit model, the landscape can be smooth yet CCTM would still be quite high.
4. The notion of smoothness is only observed from visualization and lacks any metrics. Computing the eigenvalue of Hessian would be helpful for estimating the correlation between CCTM and smoothness.
5. As shown in Table1, the recall accuracy is similar for VGG, ResNet, DLA and GoogleNet and may be the reason why they observe the same pattern in CCM. I believe it would be more convincing if the author can try a wider range of models with different complexity, including ViT.
6. Only CIFAR-10 is used in the analysis and may be an exception. Hope more multi-label classification datasets could be analyzed, e.g. MNIST, and ImageNet.

Questions:
1. Could you please provide some intuition why higher CCTM results in sharper minimum, especially when the loss space is not in the two classes involved in CCTM calculation ( as shown in Figure 4) ?
2. Is class interference possibly an inherent problem of a dataset? As stated in the paper, cats are often confused with dogs because of low resolution. It seems that training techniques like anneal-lr can alleviate this problem, but is there a lower-bound on class interference?

**Summary Of The Paper:**

The authors propose the notion of class interference represents the learning difficulty in data. They propose using a CCTM metric for understanding class interference whereas a larger interference metric indicates a sharper minimum and worse generalization. They also give an explanation for why annealed learning rate results in better generalization. The authors perform analysis on multiple architectures on CIFAR 10 to validate their claim

**Summary Of The Review:**

Overall, I believe the paper is not complete in terms of writing and experiments. Although the observations are interesting, they are not supported by enough empirical and theoretical analysis. I've changed the score to strong reject as it seems that the authors are not open to improvements and I believe this paper is quite below ICLR standard.

---

> ### Author Response · Authors · 2022-11-04
> **motivation not clear**
>
> well. This is really hard to understand. It's clear that from Fig 1 and Table 1, and the introduction that the paper is about understanding the generaliation of deep neural networks.
>
> If this message cannot be ciphered from the paper, perhaps it's a waste of time of discussing anything else.

---

> ### Author Response · Authors · 2022-11-05
> **what's the point of adding more models?**
>
> the recall accuracy is similar for VGG, ResNet, DLA and GoogleNet and may be the reason why they observe the same pattern in CCM. I believe it would be more convincing if the author can try a wider range of models with different complexity, including ViT.
>
> >>ResNet is considered the most popular networks in DL. DLA extends it in a fairly sophisticated way ( a tree hierarchy with skip connections between the trees and inside). Is ViT better than ResNet? in what sense? Has the reviewer seen any model with a better test accuracy than the models and training method reported in our paper? If you can give us a reference, we're willing to try. If the answer is "NOT", then perhaps this comment is not very helpful.
>
> >>There are numerous network models. The message of the Table is very clear. The learning difficulty is in the data. DATA.
>
> >> "may be the reason why they observe the same pattern": This does not really make much sense. What is the "reason" you mean here.

---

### Author Response · Authors · 2022-11-04
**writing reviews & dataset**

We really appreciate the reviewers for the comments.

However, we kindly make the following call:
1. Please read and make sure a good understanding of the motivation, problem, and methods is formed before writing the reviews. We understand that we all have limited time reading papers and writing reviews. To make a constructive reviewing system, it is hard and the literature is still working on it. Perhaps we all agree that reading before starting writing reviews is the only thing we can do to make a better reviewing sytem for our community.
2. DL research is not just benchmarking and running as many experiments as you can (There is a strong trend in DL favoring this kinds of "research"). Instead, gaining a careful understanding of the problem, designing reasonable experiments, and carefully analyzing the results, finding the reasons and designing another expriments to verify and dig deeper.

This is a very insightful paper. It is sad to see none of the reviewers reads much in first round review and we received three "jumpy" rejection decisions, and so far none of them understands the paper's key ideas. This may be due to our imperfect transition in writing, which is pointed by a reviewer. We provided detailed replies to this problem, and can fix it.

Update: we just fixed the transition problems and attached the CCTMs of GoogleNet and DLA to the appendix as requested.  Rev 3 had very good equestions during the feedback time and we benefit a lot from the questions.

A reading across the reviews also show that reviewers may be not sure of the literature of this research, especially about the problem, progress, the study methods and what dataset to use. This is fine becaue the DL literature is big. Nobody knows everything. We kindly ask the reviewers to take a look at the references in the introduction, especially Li. et. al. 2018 and the 2nd reference below (which was found by us after submission; we will cite it in the revision too), about the problem of minima visualization, and compare what we found, and see if we pass the ICLR standard.

As per the comments that our paper only uses the dataset of CIFAR-10, please note this is not an optimizer or architecture paper. This is a visualization of the generalization ability of deep models. For this direction, it is common to use only one dataset and in fact, CIFAR-10 is the most common one:

1. Visualizing the Loss Landscape of Neural Nets by Li, et. NeuriPS 2018
2. Can Neural Nets Learn the Same Model Twice? Investigating Reproducibility and Double Descent from the Decision Boundary Perspective. by Somepalli et.al., CVPR 2022

The two papers studied the loss contour, minima sharpness, generalization and decision boundary, and both of them used the CIFAR-10 dataset ONLY.

This is becasue
1. Computation of visualization involves the evaluation of many perturbed models, so with our method.

2. Gaining understanding of generalization via visualization is still explorative and works in this regard are very few. It is important to gain a focused understanding of related issues and phenomenons in one single dataset. For a conference paper, considering the efforts in designing the experiment setup, visualization figures, and desriptions, and verification of findings, a single dataset already poses a great study focus.

3. Understanding the CIFAR-10 better is also a good contribution since it is so widely used in the community. Cats/Dogs are easily mis-classified during Testing, and we discover that with our method in Training. Plus we found Cats and Frogs interfere each other in learning. We also have an explanation for it. Isn't that cool?

The discovery of this paper may be fundamental to understanding the learning difficulty and datasets in deep learning. Please don't miss the opportunity of giving this paper a careful read. Thank you all for your community efforts.

---

### Decision · Program_Chairs · 2023-01-20

**Decision:**

Reject

**Justification For Why Not Higher Score:**

Consensus of reviewers to reject.

**Justification For Why Not Lower Score:**

N/A

**Metareview: Summary, Strengths And Weaknesses:**

There was a consensus among the reviewers to reject this paper. They thought it lacked clarity and it had some technical issues (mistake in the proof according to reviewer aWAb). Reviewer aWAb recognized the improvements that the authors made in their revision, thus increasing slightly their score, but they still thought that the paper needed to go through another round of review.

While this is not the reason for rejection, I will add that the authors showed an utter lack of respect towards the reviewers, thereby violating part of the ICLR Code of Ethics "Researchers must show respect for colleagues (...)". This was not a constructive way to engage with the reviewers.